# Genetic and Epigenetic Interactions Involved in Senescence of Stem Cells

**DOI:** 10.3390/ijms25179708

**Published:** 2024-09-07

**Authors:** Florin Iordache, Adriana Cornelia Ionescu Petcu, Diana Mihaela Alexandru

**Affiliations:** 1Biochemistry Disciplines, Faculty of Veterinary Medicine, University of Agronomic Sciences and Veterinary Medicine, 050097 Bucharest, Romania; a.ionescu20@s.bio.unibuc.ro; 2Advanced Research Center for Innovative Materials, Products and Processes CAMPUS, Politehnica University, 060042 Bucharest, Romania; 3Pharmacology and Pharmacy Disciplines, Faculty of Veterinary Medicine, University of Agronomic Sciences and Veterinary Medicine, 050097 Bucharest, Romania; diana.alexandru@fmvb.usamv.ro

**Keywords:** stem cells, epigenetics, methylation, acetylation, histone, cellular senescence

## Abstract

Cellular senescence is a permanent condition of cell cycle arrest caused by a progressive shortening of telomeres defined as replicative senescence. Stem cells may also undergo an accelerated senescence response known as premature senescence, distinct from telomere shortening, as a response to different stress agents. Various treatment protocols have been developed based on epigenetic changes in cells throughout senescence, using different drugs and antioxidants, senolytic vaccines, or the reprogramming of somatic senescent cells using Yamanaka factors. Even with all the recent advancements, it is still unknown how different epigenetic modifications interact with genetic profiles and how other factors such as microbiota physiological conditions, psychological states, and diet influence the interaction between genetic and epigenetic pathways. The aim of this review is to highlight the new epigenetic modifications that are involved in stem cell senescence. Here, we review recent senescence-related epigenetic alterations such as DNA methylation, chromatin remodeling, histone modification, RNA modification, and non-coding RNA regulation outlining new possible targets for the therapy of aging-related diseases. The advantages and disadvantages of the animal models used in the study of cellular senescence are also briefly presented.

## 1. Introduction

Cellular senescence is a permanent condition of cell cycle arrest caused by a progressive shortening of telomeres defined as *replicative senescence*. This process is a physiological reaction to stop the accumulation of DNA damage and genomic instability, but also promotes the remodeling of tissue during development [1,2,3].

Stem cells may also undergo an accelerated senescence response, distinct from telomere shortening, known as *premature senescence* [1,4,5]. This process is produced by a primary senescent cell as a response to different stress agents like genotoxic agents, radiation (ionizing and UV), numerous drugs (bleomycin or doxorubicin induce DNA damage; abemaciclib and palbociclib block cyclin-dependent kinase CDKs), the activation of oncogenes (Ras, B-Raf), inactivation of tumor suppressors such as PTEN, epigenetic modification (DNA methylases or histone deacetylases), and paracrine signaling with the appearance of senescence-associated secretory phenotype (SASP). These mechanisms determine the accumulation of senescent cells both in tumor and normal tissue [5,6,7,8,9,10,11].

The accumulation of senescent cells can contribute to age-related diseases like osteoarthritis, atherosclerosis, pulmonary fibrosis, chronic kidney disease, diabetes, aged-related macular degeneration, neurodegenerative diseases (Parkinson’s, Alzheimer’s), myelofibrosis, and cancer [12,13,14,15,16].

There are several drug classes in clinical use for the management of senescence: metformin, spermidine, navitoclax (ABT263), venetoclax (BCL-2 inhibitor, for the treatment of myelofibrosis, idiopathic, pulmonary fibrosis), dasatinib (tyrozine kinase inhibitor, Alzheimer’s disease, kidney chronic disease, cancer), antioxidants (quercitin, hyperoside, fisetin, curcumin, luteolin, PI3K/AKT pathway inhibitors), 17-DMAG/IP1504 (HSP90 inhibitor, pulmonary fibrosis, aged-related macular degeneration), UBX0101/FOX04-DRI peptide (P53/MDM2 pathway inhibitor, osteoarthritis, pulmonary fibrosis), GMD/SSK1 (galactosidase prodrugs, lung fibrosis, cancer), cardiac glycosides (ouabain, proscillaridin A, digoxin), antibody–drug conjugates (anti-apolipoprotein D/pyrrolobenzodiazepine, anti-βeta-microglobulin/duocarmycin), and CAR-T cells therapy [17,18,19,20,21,22,23].

Recent studies involve the development of senolytic vaccines that aim to target senescent cells form tissues and from the immune system, which are responsible for age-related diseases. Other types of senolytic vaccines target the release of aging molecules and aging factors [24,25]. Although there are clinical trials involving these senolytics and their role in lowering the number of senescent cells in humans, the short-term and long-term side-effects of these therapies are still unknown [19].

The purpose of this review is to highlight the new epigenetic interactions that are involved in stem cell senescence, providing a useful update regarding recent senescence-related epigenetic alterations such as DNA methylation, chromatin remodeling, histone modification, RNA modification, and non-coding RNA regulation, outlining new possible targets for the therapy of aging-related diseases. In addition, the advantages and disadvantages of the animal models frequently used in the study of cellular senescence are summarized.

## 2. Cellular Senescence: Mechanisms and Pathways

Cellular senescence induces metabolic changes and morphological alterations such as modifications of cell size and shape, changes in the constitution of cell membrane, high number of mitochondria, increased activity of lysosomal enzymes, and nuclear modifications [26].

Stem cells are not impacted by replicative senescence, although they are nonetheless prone to harm and proliferate during the ageing process. Stem cells tend to accumulate DNA damage with ageing, which diminishes their capacity to regenerate cell lineages, resulting in age-related loss of organ function and homeostasis as well as an increase in the prevalence of age-related disorders. However, there are not many data about what causes this damage or how it contributes to the senescence of stem cells. DNA damage can cause stem cell numbers to decrease by activating apoptosis, ageing, and differentiation pathways. Research studies have also demonstrated that aberrant cell proliferation, tumor-like alterations, and reduced self-replication of stem cells are caused by elevated ROS levels in ageing mesenchymal stem cells and elevated ROS expression in mouse hematopoietic and neural stem cells. Also, autophagy dysregulation results in abnormalities related to protein homeostasis, poor protein folding, and the build-up of abnormal proteins, which can cause damage to cells and tissues as well as deplete stem cells. Reduced stem cell uptake of nutrients is caused by an increase in point mutations and deletions in the mitochondrial DNA. The regulation of stem cell function is significantly influenced by epigenetic regulation, and alterations in the epigenome with ageing impact the stem cell ageing process. DNA methyltransferases regulate the balance between differentiation and self-renewal in many adult stem cell compartments. Genetic material damage, non-coding RNA and exosomes, loss of proteostasis, intracellular signaling pathways, and mitochondrial malfunction are the hallmarks of the molecular mechanisms of stem cell ageing. A marker of cellular senescence p16INK4A accumulates with age in NSCs, HSCs, and satellite cells [26,27,28].

The senescent cells found in tissues and organs play a crucial role in both the emergence of chronic diseases and the ageing process. Called cellular senescence, ageing causes cells subjected to metabolic, genotoxic, or oncogene-induced stress to enter a nearly irreversible cell cycle stop. An increase in inflammatory mediators, primarily cytokines and chemokines, known as the SASP, is a prominent characteristic of senescent cells and is hypothesized to contribute to disease. This disruption of stem cell regeneration, tissue and wound repair, and inflammation results in harm to the dynamic equilibrium [28,29].

Cellular metabolism possesses an important role regarding the senescent cells’ function and fate. The explanation behind this role is that senescent cells have a very active but transformed metabolism, characterized by elevated glycolysis and mTOR activity, even if they do not undergo cellular division [27]. This metabolic requirement is required by their growth (cell expansion without cell division is a hallmark of senescence) [28], an increased secretion of inflammatory factors (SASP phenotype), and increased oxidative stress that can cause the accumulation of unfolded proteins (endoplasmic reticulum stress), apoptosis, necrosis, and autophagy. Furthermore, a hallmark of senescent cells is apoptosis resistance following ROS and other genotoxic exposures [29,30]. Also, the high metabolic demands of senescent cells may arise from a metabolically highly active cell that has undergone extensive changes in protein expression and production, finally developing the SASP [31]. SASP promotes inflammation, angiogenesis, cell invasion, and proliferation, explaining how senescent cells can promote aging phenotypes and related pathologies. This phenotype comprises a variety of secreted proteases, soluble factors (interleukins, growth factors, chemokines), and insoluble factors (proteins or extracellular matrix molecules) that exert influence on neighboring cells via activating cell-surface receptors and their signal transduction pathways with influences in ultimately multiple pathologies, including cancer [32]. SASP proteases possess three main roles: (1) the removal of membrane-associated proteins, leading to soluble membrane-bound receptors; (2) the degradation of molecules, which plays an important role in different signaling pathways; and (3) the remodeling of the extracellular matrix. All these factors assist senescent cells in the modification and remodeling of the tissue microenvironment, with implications in the loss of tissue homeostasis due to limited regenerative tissue capacity caused by cell cycle arrest, and altered functions of surrounding cells because of SASP, with both processes linked to the metabolic state of senescent cells [28].

Through the secretion of cytokines and chemokines (e.g., IL1α, IL1β, IL6, IL8, CXCL1, CXCL2), growth factors (e.g., amphiregulin, EGF, BMPs, FGFs, VEGF, WNTs), extracellular matrix components (e.g., fibronectin), and proteases (e.g., MMPs, plasminogen activators), as well as exosome-like small extracellular vesicles, the SASP mediates the paracrine activities of senescent cells. Cells that have aged or experienced senescence release many extracellular vehicles (EVs). Apoptotic bodies, microvesicles, and exosomes are examples of EVs that facilitate material transfer and cell-to-cell contact. MSCs release exosomes with modified payloads, like miRNAs, in large quantities as they age. SASPs can trigger the proliferation, angiogenesis, or epithelial–mesenchymal transition of neighboring or cancer cells [28,29,30].

Senescent cell size and shape display an enlarged, flattened and irregularly shaped cell body characteristic, with numerous or enlarged nuclei, due to the activation of the mTOR pathway, and an increase in cell–matrix adhesions. In senescent cells, the number and activity of membrane-bound organelles, especially mitochondria, lysosomes, and endoplasmic reticulum, are increased. Furthermore, to maintain homeostasis and to compensate for their diminished activity in response to oxidative stress, senescent cells increase the production of organelles. The newly formed organelles can also be prone to oxidative stress and an increased number of alterations, with implications in the progression of senescence phenotype [33]. Also, the reorganization of vimentin filaments, a cytoskeletal protein involved in proliferation, migration, signaling, and organelle anchorage may have a major contribution to the altered shape of senescent cells [34].

The plasma membrane plays a key role in interactions with adjacent cells and extracellular space. An important modification regarding the constituents of the senescent cells’ plasma membrane is the upregulation of caveolin-1, a plasma membrane protein implicated in the formation of caveolae (invaginations of the cell membrane that mediate the transport of liquid and other molecules) [35]. Caveolin-1 protein is upregulated by p38 (MAPK pathway), thus inducing morphological alterations and influencing the adherence of senescent cells. Furthermore, a form of oxidized vimentin was also observed in the plasma membrane of senescent fibroblast [26].

Mitochondrial dysfunctions are common in senescent cells in terms of altering ATP production and displaying impaired mitochondrial oxidative phosphorylation and diminished inner membrane potential, followed by an overproduction of reactive oxygen species. The incapacity to maintain the redox balance lowers the cells’ capacity to adapt to stress conditions and promotes senescence through signaling pathways. Also, a reduction in ATP production lowers the AMP-ATP ratios, which activates 5’ AMP-activated protein kinase (AMPK), promoting the mitochondrial activation of the autophagy catabolic pathway [36]. Moreover, AMPK induces cell cycle arrest through the phosphorylation of p53 that controls the cells’ response to stress [37] and limit glycolysis in senescent cells by activating fatty acid oxidation, tricarboxylic acid cycle, and oxidative phosphorylation [38]. In response to this perturbation, cells improve ATP production by increasing the glycolysis to compensate the ATP decrease. On the other hand, the number of mitochondria in senescent cells cannot be controlled, considering that dysfunctional mitochondria cannot be removed through mitophagy (a type of autophagy that removes damaged mitochondria during metabolic stress) [39], thus maintaining the cellular senescence. This perturbation contributes to mitochondrial accumulation and the installation of senescence-associated mitochondrial dysfunction (SAMD) [40].

Lysosomal proteins are overexpressed; therefore, an expanded lysosomal content is a hallmark of senescence. Lysosomes are organelles with an acidic pH, characterized by many hydrolytic enzymes implicated in protein degradation. In cells that undergo senescence, the activity of some enzymes increases, such as α-mannosidase, α-fucosidase, N-acetyl-β-hexosaminidase, and lysosomal β-galactosidase (used as a senescence marker, even if it lacks specificity) [41,42].

Endoplasmic reticulum expansion is also a hallmark of senescence, maintaining a high synthesis and secretion rate of inflammation factors related to SASP. A cause of the increase rate of translation is the accumulation of the unfolded or misfolded protein levels that can generate cellular stress in the endoplasmic reticulum [43]. Consequently, the number of reactive oxygen species (ROS) increase, thus affecting protein folding and the formation of disulfide bridges in proteins secreted in SASP. Furthermore, the abnormally folded protein can aggregate, leading to a loss of cellular homeostasis or proteostasis. Therefore, the cellular response limits endoplasmic reticulum stress by the inhibition of protein translation, structural reorganization of misfolded proteins, and even their degradation via proteasome [44].

Nuclear changes are also common marks of senescent cells. For example, the downregulation of lamin B1, a protein of the nuclear envelope also known as an intermediate filament protein, destabilizes the nuclear integrity, inducing the constitutive heterochromatin loss and leading to gene expression changes [45]. A significant contribution in maintaining the constitutive heterochromatin silenced have *senescence-associated heterochromatin foci* (SAHF), that are specific regions of facultative heterochromatin. Hence, SAHF can also be used as marker of senescence associated with cell cycle arrest [46] (Figure 1).

Senescence is also characterized by DNA damage, which induces the activation of canonical senescence-related signaling pathways, including p53/p21WAF1/CIP and p16INK4A/pRb [47]. p16 provides two pathways across other tumor suppressor genes that define the intrinsic branch of cellular senescence: the p16–pRB pathway, in which p16 regulates pRb (retinoblastoma protein) upstream, and the p53–p21 pathway, in which p21 functions as p53′s downstream effector. Inhibiting the cyclin-dependent kinases CDK4/6 and CDK2 is the primary regulatory step for these pathways, as it governs the cell cycle and links them to aging, tumorigenesis, tissue repair, cellular senescence, and growth. p16 overexpression is not always linked to elevated SASP, aging, or senescence and it could be indicative of cell cycle inhibition [48]. Both p53/p21WAF1/CIP and p16INK4A/pRb tumor suppressor signaling pathways mediate the cell cycle arrest in senescent cells through the activation of p53 and p21 and the inhibition of pRb phosphorylation by cyclin-dependent kinases [49]. Also, they support the senescence program by regulating the expression of p53 and pRb transcriptional factors, considering that they are inhibitors of cyclin-dependent kinases, downregulating cell cycle progression factors [50]. pRb is a tumor suppressor gene that restricts cell cycle progression through interaction with histone deacetylases leading to the repression of genes and preventing uncontrolled cellular growth. It is activated by p16INK4A in response to DNA damage and inactivated through phosphorylation by cyclin-dependent kinases (mono-phosphorylation, which generates a metabolic increase and the accumulation of hyperphosphorylated isoforms, resulting in irreversible uncontrolled cellular division) [51]. If pRb is mutated, senescent cells undergo apoptosis to restart DNA synthesis. Through cyclin-dependent kinase inhibition, p16INK4A functions as a positive regulator of pRb and is essential in the formation of SAHF. Unsurprisingly, p16INK4A is usually inactivated in human tumors that have already progressed and functions as a tumor suppressor [52]. Moreover, p16INK4A accumulates in primary human cells during replication and oncogene-induced senescence, while p16INK4A expression initiates the senescence program that includes cell cycle arrest, morphological senescence-associated alterations, increased SA-β-Gal activity, and the appearance of SAHF [53]. On the other hand, the p53/p21WAF1/CIP signaling pathway is triggered as a consequence of DNA damage due to oxidative stress, oncogenic transformation, or telomere shortening. The continuous activation of p53 caused by constant DNA damage response (DDR) signaling results in cellular senescence [54]. Numerous post-translational changes, including phosphorylation, methylation, and acetylation, are necessary for the activation of p53 [55] (Figure 1). p21WAF1/CIP is encoded by the CDKN1A gene, part of the Cip/Kip family of cyclin-dependent kinase inhibitors (CDKIs), besides p27 and p57, associated with CDK inactivation and thus preventing cell cycle progression. Through its interactions with cyclins via the two cyclin binding pockets (Cy1 and Cy2), it suppresses the kinase activity of CDK complexes. This results in the blockage of Rb family phosphorylation, causing cell cycle arrest [47,56]. Increased levels of p21WAF1/CIP limit the kinase activity of cyclin D/CDK4,6 complexes, restricting cell cycle progression, while a decreased expression of p21WAF1/CIP activates the cyclin D/CDK4,6 complex, promoting cell cycle progression. In some cancers, like glioblastoma, therapeutics can induce more senescence than cell death, indicating that the senescence pathways can be easily evoked, which has significant therapeutic implications [30].

Also, p21WAF1/CIP was the first transcriptional target for p53. Unlike the INK4 family of CDKIs that particularly bind and inhibit CDK4 and CDK6, thus triggering the arrest of cell cycle progression during the G0/G1 phase, p21WAF1/CIP triggers cell cycle arrest at all stages of the cellular cycle due to its interaction and inactivation of various cyclin/CDK complexes. Prolonged p53 expression and p16INK4A activation caused by additional stimuli and ongoing stress can lead to continuous cell cycle arrest, influencing cell fate [47,57] (Figure 1).

## 3. DNA Methylation Involved in Stem Cell Senescence

To generate sufficient material for fundamental study or cellular therapy, stem cells are multiplied several times in vitro. However, cell culture expansion has a significant and ongoing effect on stem cell growth, phenotype, gene expression, proteomics, and physiological function until they eventually enter in a state called replicative senescence [58,59]. DNA methylation level can be used for the estimation of passage numbers and cumulative population doublings, as demonstrated by Franzen et al. (2021) who established the limit at only six cytosine/guanine dinucleotides (CpG sites) [60].

The fifth carbon of cytosines (5-methylcytosine, or 5mC), which is primarily present in symmetrical CpG dinucleotides, is methylated in bacteria and eukaryotes [61]. Regardless of its ancient origins, DNA cytosine methylation no longer occurs in several eukaryotic organisms such as *Caenorhabditis elegans, Drosophila melanogaster*, fission yeasts, and bakers’ yeasts that display virtually no 5mC. The cost of the methylation process is represented by the fact that 5mC can be easily mutated because it can spontaneously undergo deamination, leading to cytosine → thymine transitions [62]. Only symmetrical CpG methylation is preserved during DNA replication, although de novo DNA methylation can happen in any sequence context. Furthermore, DNA methyltransferases (DNMTs) add toxic 3-methylcytosine lesions into DNA, but the coevolution with specific alkylation repair enzyme (ALKBH2) permits eukaryotes to endure DNA methylation. Mammalian genomes have exceptionally high levels of CpG methylation; 70–80% of CpGs are methylated, with some variations, depending on the tissue. ALKBH2 and proliferating cell nuclear antigen (PCNA) colocalize in DNA replication foci; however, it is unclear if these two proteins can interact alone or need other molecules [63].

DNA methylation and demethylation factors. DNMT3A and DNMT3B are the two main de novo DNA methylation enzymes found in mammals. These enzymes contain a highly conserved carboxy terminal DNMT domain (MT-ase domain) and two chromatin reading domains: ATRX-DNMT3-DNMT3L (ADD) and PWWP. Additionally, DNMT3L, a catalytically inactive DNMT, interacts with DNMT3A and DNMT3B in the germline and specifically increases their activity.

A maintenance methylation enzyme, DMNT1 is involved in the symmetrical DNA methylation at the replication process (Figure 2a, Table 1) [64].

At the promoter level, DNA methylation does not occur in trimethylated histone H3 Lys4 (H3K4me3); the ADD domain binds only to unmethylated H3K4, forming the MTase domain and enabling DNA methylation (Figure 2b) [65].

In mouse embryonic stem cells (ESCs), DNMT3B protein with a PWWP mutation lacks affinity for H3K36me3-marked gene bodies, and in Setd2 mutants, DNMT3B is absent from gene bodies (Figure 2c) [64,65].

At the replication level, UHRF1 (E3 ubiquitin-protein ligase) specifically binds hemimethylated CpG dinucleotides through its SET- and RING-associated (SRA) domain to H3K9me2 and H3K9me3 through its tandem TUDOR-PHD (TTD) domain. UHRF1 recruits DNMT1 through its ubiquitin-like (UBL) domain, allowing the RFTS domain of DNMT1 to bind to histone H3 tails that were formerly ubiquitylated by the RING finger domain of UHRF1. Additionally, MTase-domain of DNMT1 methylates the daughter DNA strand (Table 1, Figure 3) [64,66].

DNA demethylation is catalyzed by the TET methylcytosine dioxygenases (ten-eleven translocation (Tet) enzymes), which oxidize 5mC in steps following 5-hydroxymethylcytosine (5hmC), 5-formylcytosine (5fC), and 5-carboxylcytosine (5caC). Demethylation can also occur by thymine DNA glycosylase (TDG) that removes a base followed by the activity of the base excision repairing pathway [67,68,69].

Stem cell pluripotency factors KLF4, OCT4, and HOXB13, NKX, neural patterning factors, and C/eBPα use a methyl-specific binding motif, which is involved in cell-type transitions, or are otherwise refractory to transcription activation [70,71,72,73]. Using knockout mice for adapter proteins for site-specific demethylation by TET methylcytosine dioxygenases Gadd45a and Ing1, Schafer et al. (2018) showed that impaired DNA demethylation of C/EBP sites caused premature aging. They concluded that double-knockout mice manifest the premature appearance of some (adipogenic differentiation, elevated cytokine level and inflammation, reduced body weight and lipodystrophy, female infertility), but not all the characteristics observed in natural aging, such as a condition known as segmental progeria [72].

Sakaki et al. (2017) showed that replicative senescent cells have different DNA methylation patterns compared to in proliferative stem cells. Particularly, immune response gene-related CpG sites showed enhanced hypomethylation. Remarkably, few alterations in methylation patterns were noted in cases of senescence due to Ras-induced oncogenic stress. Additionally, immune response-related genes’ non-CpG island promoters were shown to enrich hypomethylation, which may have an impact on the activation of specific SASP genes [74].

Tet enzymes have a role in the ageing of the hematopoietic and reproductive systems, according to recent data. Tet1 deficiency lowers fertility and causes premature ovarian insufficiency or accelerated reproductive failure syndrome. Tet1-deficient animals experience an accelerated age-related decline in fertility due to a gradual loss of spermatogonia stem cells and spermatogenesis. Somatic Tet2 mutations enhance the chance of developing hematological malignancies, such as acute myeloid leukemia and myeloproliferative and myelodysplastic neoplasms, as well as myeloid expansion and innate immune dysregulation with ageing [75,76,77].

Based on next-generation sequencing techniques, an “epigenetic clock” was postulated and explains highly repeatable methylation patterns that precisely predict a person’s chronological age. While there is a lot of overlap between senescence and age-related changes in DNA methylation, there are also clear distinctions at CpG sites, most notably those involving Hox genes and genes that regulate cellular differentiation. The discovered commonalities imply that comparable processes govern the regulation of both senescence and ageing. Previous studies have also shown that the global methylation profiles of cancer cells and senescent cells are very similar. Nonetheless, a more recent examination of the methylation of senescent cells revealed important regional variations in methylation events between transformation and replicative senescence. Indeed, promoter regions of genes controlling metabolism and cellular biosynthesis were revealed to have CpGs methylated in senescent cells. On the other hand, altered cells showed stochastic methylation of genes controlling cell growth and differentiation, which may promote self-renewal and prevent differentiation [78,79,80].

Researchers used an mRNA cocktail that expressed OSKMLN, the pluripotent stem cell (iPSC) reprogramming factors, to transfect aged primary human fibroblasts and endothelial cells. With this approach, epigenetic reprogramming is possible without changing the original cell identity. Following therapy, the epigenetic clock methylation pattern of aged primary cells rapidly reverted, giving them a youthful appearance. Further immunofluorescence data revealed a young epigenetic profile with enhanced nuclear staining of HP1, H3K9me3, and the nuclear lamin support protein lamin-associated polypeptide 2 (LAP2) [81,82].

A mutation in the lamin A gene was present in the LAKI progeroid mouse model, which was also reprogrammed in vivo. H3K9me3 and H4K20me3 heterochromatin maintenance markers were returned to healthy levels by the short-term cyclic expression of OSKMLN. When compared to control mice, the liver cells extracted from these mice contained a notably lower number of SA-β-Gal positive cells. Moreover, fibroblasts taken from these progeroid animals that were reprogrammed showed improved mitochondrial function, reduced DNA damage, and decreased cellular senescence as indicated by the production of β-galactosidase and Il-6. An elderly mouse glaucoma model’s lost eyesight was recovered, and greater axon regeneration following injury resulted from the use of this transfection technique to restore youthful DNA methylation patterns and transcriptomes in the central nervous system tissue of aged mice [80,81,82].

Mixed-lineage leukemia 1 (MLL1), an H3K4 methyltransferase, is necessary for SASP stimulation since silencing MLL1 with shRNA significantly reduced the level of several SASP genes (MMP1, MMP3, IL-1A, IL-6, IL-8) in oncogenic induced senescence (OIS)-transformed fibroblasts and OIS-primary human melanocytes. The production of SASP proteins in response to DNA damage did not cause senescence and it was interestingly suppressed by MLL1, suggesting an anti-inflammatory function, independent of senescence [83].

Both DNMT and Tet enzymes are expressed differently in senescent hematopoietic stem cells (HSCs), and animals lacking these alleles exhibit traits associated with aged HSCs, including myeloid skewing and a propensity for cancer. Additionally, epigenetic regulators like Tet2, DNMT3A, and ASXL1 are the most altered genes in hematopoietic cells throughout aging [84].

Researchers looking at the levels of DNA methylation in populations of young and old mouse HSCs have found significant variations in DNA methylation that appear to be locus-specific and include both hypomethylated and hypermethylated DNA. It is interesting to note that areas with hypermethylation also overlap with histone H3K27 methylation, or classical repressor marks, and PRC2 (polycomb repressor complex 2)-genome-rich regions, indicating a close relationship between these two marks to maintain gene repression [85].

Data comparing HSCs generated from young and senescent cells using whole-genome bisulfite sequencing, revealed that 70% of the genes involved in stem cell maintenance showed hypomethylation in senescent cells, related with an increased expression of those genes. The transcription factor PU.1, a crucial regulator of HSC development, has binding sites that are marginally correlated with hypermethylation, according to the same study. These findings support the idea that in senescence, modifications in DNA methylation promote HSC renewal and prevent differentiation. Furthermore, it was discovered that ribosomal genes exhibit hypomethylation and are frequently targeted during the aging process [86]. Genome-wide investigations showed that numerous genes have been found to be regulated in senescence. For example, lysine-specific demethylases (kdm3a-b, kdm5-d, and kdm6a-b), which have been shown to have roles in stem cell biology, have lower levels during senescence. One demethylase that promotes differentiation by suppressing genes involved in HSC self-renewal is KDM5B, an H3K4 demethylase that is generally highly expressed in primitive HSCs. The lack of differentiation and enlarged HSC compartment seen in elderly mice may be partially explained by the age-related decrease in KDM5B, which may also lead to elevated levels of H3K4me2 and H3K4me3 in HSCs [87,88].

The chromosomal distribution of 5hmC in senescent mesenchymal stem cells (MSCs) has been investigated recently. Based on these findings, 134.693 hydroxymethylated CpG sites were found in people between 2 and 89 years old. The majority of the 5hmC sites were found in exons, with a lower amount found in promoters, 5′UTR, and introns. Interestingly, its relationship with poised enhancers was demonstrated by a significant association with the H3K4me1 enhancer marker, but not with H3K27Ac [88]. In senescent MSCs, differently hydroxymethylated CpG sites were found, with 48% of them gaining 5hmC, while 52% lost it. Gene ontology analysis revealed that development-specific genes were enriched in both hyper- and hypohydroxymethylated DNA. Morphogenesis was linked to hyperhydroxymethylation, while differentiation was linked to hypohydroxymethylation. Furthermore, genomic areas with hypomethylated DNA coincided with hyperhydroxymethylated regions [84,89].

## 4. Histone Modification and Chromatin Remodeling Complexes Involved in Stem Cell Senescence

### 4.1. Histone Methylation

Histone modifications come in several forms, such as methylation, acetylation, phosphorylation, ubiquitination, ADP ribosylation, and proline isomerization. Depending on how many lysine residues are methylated, histone methylation is classified as mono- (me1), di- (me2), or trimethylated (me3). Different locations of histone methylation are connected to the activation or repression of genes. Senescence-related alterations in H3K9me3, H4K20me3, H3K27me3, and H3K9ac levels have been documented both in vitro and in vivo studies. These modifications are regulated by an exclusive number of enzymes that maintain a balance in adding and removing these functional groups.

***H3K4me3*** levels are regulated by histone demethylases KDM5A and ASH1L. H3K4me3 enrichment in senescence-related gene promoter regions was observed by researchers using somatic cells in *C. elegans* model, this dynamic frequently takes place in areas with relatively low H3K4me3 markers. In contrast, inhibition of the ASH-2 Trithorax complex results in H3K4me3 deficiency and extension of lifespan [90,91]. In a mouse model of Alzheimer’s, the level of H3K4me3 and its catalytic enzymes is increased in the prefrontal cortex. Treatment with different inhibitors of H3K4 histone methyltransferase boosts the recovery of prefrontal cortex function of mice [92]. Studies on mouse hematopoietic stem cells demonstrated a similar pattern of H3K4me3 [93]. However, a recent study employing human senescent HSCs showed that senescence is linked to decreased levels of H3K4me3. Knockdown of Ash1L reduces the H3K4me3 level at the Hoxa10, Osx, Runx2, and Sox9 promoters.

***H3K27me3*** is typically linked to gene silencing and compacted heterochromatin. Senescent cells from patients with Hutchinson–Guildford progeroid syndrome (HGPS) and from *C. elegans* were found to have lost global H3K27me3, while in killifish and mouse brains, the global level of H3K27me3 increased. The histone lysine demethylases jmjd-1.2/PHF8 and jmjd-3.1/JMJD3 may function as positive lifespan regulators in response to mitochondrial dysfunction in a variety of species, indicating that lifespan is negatively impacted by elevated levels of H3K27me3 in genes related to the mitochondrial unfolded protein response. Uncovering the influence of H3K27me3-modifying enzymes on aging will require identifying the cell-type-specific roles of H3K27me3 in lifespan regulation [94].

Jing et al. (2016) showed that H3K27me3 levels in bone marrow senescent MSCs were significantly increased compared to the MSCs from normal mice [95]. Also, the same results were obtained in another research by Li et al.l (2017) who reported that H3K27me3 was associated with senescent bone marrow MSCs [96]. The levels of H3K27me3 are regulated by histone methyltransferases: EZH2, KDM6A (UTX), and KDM6B (JMJD3). In the promoter regions of cell cycle inhibitor genes P15INK4b, P16INK4a, P21CIP1, and P27KIP1, EZH2 increases H3K27me3.

KDM6A and KDM6B are histone demethylases intended to reduce di- and trimethylation on H3K27. KDM6B is significantly reduced in senescent MSCs from old mice compared to young mice. Knockdown of KDM6B decreased bone marrow MSC-mediated bone formation via increasing H3K27me3 levels at the promoters of BMP2, BMP4, and HOXC6–1 [97].

***H3K9me3*** and ***H3K36me3*** are also crucial for senescence. The histone demethylase KDM4B, often referred to as JMJD2B, has catalytic activity against the following histone residues: H1.4K26me3, H3K9me3, H3K9me2, H4K20me2, H3K36me3, and H3K36me2. And SETD2 (SET domain-containing 2) is the methyltransferase in control of H3 lysine 36 trimethylation. A reduced lifetime was linked to H3K36me3 deficit in *S. cerevisiae* and *C. elegans*. In mesenchymal stem cells carrying mutations of HGPS or Werner syndrome (WS) cells, aging was shown to have reduced levels of H3K9me3 and HP199.

In human and mouse senescent HSCs, the expression of H3K9me3 methyltransferase UV39H1 declines, which results in a general decrease in H3K9 trimethylation and disrupted heterochromatin function. Vitamin C, gallic acid, or low-dose chloroquine combined with Werner syndrome-specific MSC improves a variety of senescent characteristics, encouraging cell self-renewal, and increasing levels of heterochromatin-associated markers, such as H3K9me3 [98,99,100]. The maintenance of stem cell self-renewal and proliferation depends on the state of H3K9me2. MSC depletion is encouraged by downregulating KDM4B while MSC self-renewal is hampered. H3K9 alterations may influence genes implicated in signaling pathways, cell cycle processes, and cytokine-linked pathways, according to a genome-wide study involving the histone H3K9me2 patterns at different gene promoters in bone marrow MSCs. Additionally, in mouse MSCs, KDM4B was found on the promoters of the Wnt target genes, Runx2 and Ccnd1. H3K9me3 levels on their promoters were considerably elevated upon deletion of KDM4B, respectively [101].

***H3R17me*** is methylated at the arginine residues (2, 17, and 26) by the CARM1 enzyme, the primary target being arginine in position 17. When cultured in vitro, the late-passage human bone marrow MSCs showed phenotypic alterations linked to cellular senescence, in contrast to their early passage counterparts. DDR2 downregulation and cellular senescence were present in human bone marrow MSCs by inhibiting CARM1 expression [102].

### 4.2. Histone Acetylation/Deacetylation

Active gene regions have higher levels of histone acetylation, mediated by histone acetyltransferases (HATs). Along with HATs, histone deacetylases (HDACs) are thought to act as corepressors and are essential for lifespan (Table 2). Class III HDACs called sirtuins (SIRTs) improve genome stability and control lysine residue deacetylation in a way that is dependent on NAD+ levels. In vitro cultivation of human fetal placental MSCs forces them to progressively enter a senescence state. Following extended culture, there is an elevated increase in the expression of HDAC4, HDAC5, and HDAC6, as well as a considerable decrease in the acetylation of both histone H3 and H4. At passage 8 of in vitro culture, histone acetylation at the promoters of OCT4, SOX2, and TERT was lower than in cells at passage 3 [103].

After human umbilical cord blood and adipose tissue-derived MSCs underwent replicative senescence in vitro, the expression levels of c-MYC, PcGs, HDAC1, and HDAC2 were downregulated. Inhibition of HDAC1 and HDAC2 promotes H3 acetylation at the KDM6B promotor. This modification leads to upregulation of KDM6B expression, and a decreased expression of H3K27me3 on the P16INK4A promoter region associated with an increase in P16INK4A expression [104]. HDAC inhibitors block the proliferation of MSCs derived from adipose tissue or umbilical cord blood by activating the transcription P21CIP1 ⁄ WAF1 and increasing the acetylation of histone H3 and H4, which eventually block the cell cycle at the G2/M phase (Table 2) [105].

SIRT6 is a NAD^+^-dependent H3K9 histone deacetylase that controls telomeric chromatin. SIRT6 prevents senescence, since its overexpression prolongs the life of human and rat nucleus pulposus cells. The major substrate for SIRT6 seems to be H3K56. SIRT6 deficiency leads to the accumulation of ROS, which increases the vulnerability of cells to oxidative injury, leading to the impairment of the differentiation potential of MSCs. Knockout of SIRT6 upregulates P16 and P21 proteins and speeds up cellular senescence in human MSCs. SIRT6 is involved in the deacetylation of H3K56ac at the HO-1 promoter that recruit the RNA polymerase II for transcription process in MSCs [106].

HDAC4, a class II deacetylase, is polyubiquitylated and destroyed during all stages of senescence. HDAC4 selectively binds to H3K27ac at specific enhancers, such as enhancers of VEGF and AKR1E2. The treatment of HDAC4 with P300 inhibitor rescued senescence in HDAC4-depleted cells. It was determined that P300 therapy was the main cause of the senescent phenotypes in MSCs because it dramatically raised the levels of H3K122ac and H3K27ac at the proximal senescence-specific gene promoters. Replication-induced senescence can be postponed, and senescence genes downregulated just by depleting p300. The high level ok H3K4me1 displayed in senescent MSCs at the SASP-associated super-enhancer loci was correlated with a high level of H3K27ac and BRD4. HDAC9 reduced the differentiation potential of senescent MSCs partially due to impaired autophagy. Senescent MSCs presented fewer phagosomes than young MSCs and the autophagy-related proteins were reduced, suggesting impaired autophagic activity in senescent cells. Deacetylation of autophagy-related genes, such as LC3a, LC3b, ATG7 and BECN1, by HDAC9 significantly increased senescent MSCs [105].

Histone H3K14ac is reduced, and human mesenchymal precursor cell senescence is lessened when KAT7 is inactivated. In yeast and Drosophila, H3K18ac and H3K56ac are negative indicators of senescence [106].

GCN5 (general control non-depressible 5, KAT2A), PCAF (P300/CBP-associated factor, KAT2B), and HDAC9 can affect the amount of histone H3K9 acetylation. H3K9 acetylation has been shown to be lower in senescent bone marrow MSCs. Except for H3K14, most lysine sites in histone H3 and H2B are probably acetylated by GCN5. The main target of GCN5 activity in vivo is histone H3K9. In a study, GCN5 was markedly reduced in the bone marrow MSCs of elderly mice [107].

PCAF is proficient in acetylating both histone H1 and H3. Diminished PCAF binding to the promoters of BMPR1B, RUNX2 BMP2, and BMP4 proteins were observed upon PCAF knockdown and was correlated with a reduction in H3K9 acetylation. PCAF due to implications in BMP signaling was investigated for MSC-mediated bone production both in vitro and in vivo and has been shown to possibly play a role in osteoporosis [105].

### 4.3. Histone Phosphorylation and Ubiquitination

Numerous studies showed that histone H3 is phosphorylated at Ser10 (H3S10p) with the implication of Aurora kinase proteins, knockdown of Aurora B removes H3Ser10 phosphorylation and adversely affects chromosome morphology [108].

The phosphorylation level of threonine 11 in histone H3 (H3T11p) is increased under nutritional stress, when stress-responsive genes involved in metabolic transition are activated. Also, CK2 is necessary for the phosphorylation of H3T11p under stress conditions. All these data suggest that H3T11p can be used as a marker of nutritional stress and senescence. In Drosophila, histone H3S28p phosphorylation was linked to longer life expectancy and stronger resilience to oxidative stress and malnutrition [109].

The phosphorylation of histone variant H2A(X) is an important alteration that plays a key role in DNA damage response. In human cells, this modification occurs on serine 139 of the H2AX, also known as γH2AX. This phosphorylation is implicated in DNA damage response pathways and happens during all stages of the cell cycle. Protein kinases ATM and ATR carry out this phosphorylation. H3 acetylation is associated with the phosphorylation of H3S10, T11, and S28, which is highly significant in transcription activation [110].

The trimethylation of H3K4 and H3K79 requires monoubiquitination of H2B. Replicative senescent stem cells showed an interesting pattern where H2B monoubiquitination accumulated at telomere-proximal areas, concomitant with an increase in H3K4me3, H3K79me3, and H4K16ac. In general, an increase in H2B ubiquitination seems to be correlated to replicative senescence; however, the role of H2B ubiquitination in aging is yet to be explored [111,112]. Knockdown Brap (BRCA1-associated protein), a protein essential for neurogenesis, results in DNA breaks and an increase in histone H2A mono- and polyubiquitination (H2Aub). These defects are correlated with cellular senescence through proteasome-mediated H2A proteolysis, suggesting that chromatin aberrations mediated by H2Aub may be involved in multiple aging hallmarks and tissue degeneration [113]. BRCA1/BARD1 is a E3 ligase implicated in macroH2A1 K123 ubiquitination, and it was shown by Kim et al. (2017) that primary human fibroblasts with a ubiquitination-deficient macroH2A1 were defective in replicative senescence compared to their wild-type cells [114].

### 4.4. Chromatin Remodeling Complexes

The chromatin remodeling process comprises ATP-dependent chromatin remodeling complexes and histone-modifying enzymes. In mice, senescent MSCs exhibit significantly lower levels of overall acetylation of histone H3-H4 and higher amounts of H3K27me3, compared to young MSCs, thus resulting in a lower transcriptional rate and a loss of chromatin accessibility. To revitalize senescent MSCs, the chromatin structure must be remodeled, and cytoplasmic acetyl-CoA levels must be restored. Chromatin remodeling complexes can be grouped in several families: SWI/SNF, CHD, ISWI, Tip60, INO80, SIN3, SAGA, and NuRD families (Table 3) [115,116].

SWI/SNF complex is required for the activation of nutrient-responsive genes and the regulation of transcription by promoting a more open chromatin configuration. This complex is required for the co-expression of the proteins TRF1 and TRF2 that bind to telomere and contribute to maintain the telomere length and structure [115].

Knockdown of the ARID1B, a component of the SWI/SNF complex, prevents oncogene-induced senescence. ARID1B regulates p16INK4a and p21CIP1a transcription and controls DNA damage, p53 activation, and oxidative stress, suggesting that SWI/SNF uses different mechanisms to control senescence. ENTPD7, an enzyme that hydrolyzes nucleotides, is also involved in DNA damage, oxidative stress, and senescence. The expression of ENTPD7 or inhibition of nucleotide synthesis in ARID1B-depleted cells results in the re-establishment of senescence [116].

According to Grossi et al. (2020), SWINGN, a lncRNA that only interacts with SMARCB1 under proliferating circumstances, has a pro-oncogenic effect on some tumor types. The GAS6 oncogene’s activation is modulated by SWINGN within a topologically organized area, indicating that SWINGN affects the capacity of SWI/SNF complexes to induce the epigenetic activation of promoters [117,118,119,120,121].

BAF180 (PBRM1, a subunit of the SWI/SNF complex) deletion resulted in elevated p21 expression in both mouse embryonic fibroblast and hematopoietic stem cells.

NuRD remodelers from the CHD family are involved in maintaining the histone composition and are susceptible to genotoxic stress. HDAC1 physically interacts with lamin A and possibly regulates the NURD complex. Catalytic subunits of the CAF-1 and the PRC2 complexes, p150 and EZH2, are unaffected in Hutchinson–Gilford progeria syndrome (HGPS) cells. Pegoraro et al. (2009) demonstrated that NURD constituents were lost in HGPS and normally senescent cells and sustained that the integrity of the NURD complex is important for physiological development, without premature aging [121].

## 5. RNA Modification Involved in Stem Cell Senescence

While the relationship between RNA modifications and organismal ageing has not been thoroughly studied, a growing number of evidence indicates that these post-transcriptional regulatory mechanisms play a crucial role in the process of cellular senescence, the main cause of ageing, and aging-related diseases (Table 3) [118,119].

### 5.1. m6A Modification

It has been established that (N6-methyladenosine), one of the most well-researched mRNA modifications in mammalian cells, is implicated in cellular senescence. m6A modification is carried out by the following:“writer” m6A methyltransferases: METTL3/14/16, RBM15/15B, ZC3H3, VIRMA, CBLL1, WTAP, and KIAA1429;“eraser” demethylases: FTO and ALKBH5;“reader” m6A-binding proteins YTHDF1/2/3, YTHDC1/2 IGF2BP1/2/3, and HNRNPA2B1.

METTL3/14 heterodimer facilitates methylation at the 3ʹ-UTR of CDKN1A mRNAs and promote p21 translation, being the first reported m6A modification involved in senescence. The expression level of METTL3/14 and p21 is increased in oxidative-stress-induced senescence [118].

The writer METTL14 produces m6A modification that is associated with senescence. METTL14 can be induced by TNF-α, which leads to the synthesis of miR-34a-5p that promotes cellular senescence by targeting Sirtuin-1. WTAP (Wilms tumor 1-associating protein) eraser enhances m6A in the lncRNA NORAD, contributing to the interruption of the NORAD/PUMILO/E2F3 axis and speeds up senescence. In MSCs presenting pathogenic mutations of (HGPS) or Werner syndrome (WS), the MIS12 is downregulated. Knockdown of DNMT2 in mouse embryonic cells decreases the m6A level and accelerates senescence [118].

The main erasers of m6A, such as fat mass and obesity-associated protein (FTO), and alkB homolog 5 (ALKBH5) are also involved in senescence.

The ALKBH5 level is increased in senescent stem cells due to the ability to remove m6A from the DNMT3B mRNAs, which block the expression of the E4F1 transcription factor by methylating CpG islands at its promoter region and accelerating senescence.

The reader YTHDF family are also important in cellular senescence by disrupting targeted mRNAs. In a model of mouse embryonic stem cells, m6A-modified intracisternal A-particle (IAP) mRNAs recruit YTHDFs and shorten their half-life, repressing endogenous retroviruses (ERVs). The accumulation of IAP mRNAs after the deletion of YTHDFs leads to high ERV activity, resulting in senescence and diseases [118,119].

### 5.2. m5C Modification

The m5C (5-methylcytosine) modification of RNAs is also associated with senescence through the presence of tRNA methyltransferase NSUN2 (NOP2/Sun domain family). The 3ʹ-UTR of cyclin-dependent kinase 1 (CDK1) mRNAs and the 5ʹ-UTR of the CDK inhibitor p27 mRNAs are targets of NSUN2, and m5C modification facilitates the translation of CDK1 while repressing p27, both of which improve replicative senescence [120].

### 5.3. Adenosine-to-Inosine (a-to-I) RNA Editing

The post-transcriptional modification known as adenosine-to-inosine (A-to-I) editing takes place in RNA, mainly in non-coding secondary structure. The enzymes involved in this process are called adenosine deaminase acting on RNA (ADAR) proteins, ADAR1 and ADAR2. These enzymes catalyze A-to-I editing at millions of sites in humans and between 50,000 and 100,000 sites in mice. In mammals, the main function of editing by ADAR1 that is constant across species, is to inhibit innate immune activation caused by endogenous RNA produced from unedited cells. Recently Hao et al. (2022) demonstrated that ADAR1 suppresses senescence by regulating p16INK4a expression through an RNA-editing independent non-canonical pathway. The lysosomal-mediated autophagy process downregulates ADAR1 in senescent cells, which diminishes the interaction between HuR and SIRT1 mRNA and hence lowers SIRT1 mRNA stability. SIRT1 expression reduction will further encourage EIF3a’s interaction with p16INK4a mRNA, accelerating the translation of that sequence. By adjusting p16INK4a levels, this A-to-I independent route seems to be a major regulator of cell senescence [121].

### 5.4. Non-coding RNAs

#### 5.4.1. miRNAs

Most of the studies of ncRNAs in senescence involve microRNAs (miRNAs), long non-coding RNAs (lncRNAs), circular RNAs (circRNAs), R-loops, Piwi RNAs (piRNAs), and small nucleolar RNAs (snoRNAs).

Using an 863-miRNA microarray, researchers found that in senescent vs. younger cells, 64 miRNAs (miR-30d, miR-320d, and miR-339-5p) were upregulated while 16 miRNAs (miR-103, miR-107, miR-24, and miR-130a) were downregulated [122].

MiR-17 family members are known to directly target p21, hence acting as cell cycle regulators [70]. It has been discovered that the miR-17 family members miR-93 and miR-20a diminish senescent MSCs and target p21 [122].

Okada et al. (2016) found that senescent MSCs have higher levels of miR-195. The 3′UTR of SIRT1 and TERT are direct targets of this miRNA. SIRT1 is a p53 deacetylation regulator that has been shown to prevent senescence in a variety of cell types [123].

Kim et al. (2012) observed that aged MSCs generated from adipose tissue exhibited an increase in miR-486-5p. Also, stress-induced senescent chondrocytes and osteoarthritic cartilage exhibited upregulation of miRNA-204 [124].

BMI1, a regulator of p16INK4a, is the target of miR-495, which also causes MSC senescence as indicated by enhanced p16INK4a, p53, and p21 gene expressions as well as increased SA -β-galactosidase activity [125].

Senescent MSCs have higher amounts of miR-141-3p that target ZMPSTE24, an enzyme that cleaves prelamin A to produce mature lamin. Therefore, miR-141-3p blocks ZMPSTE24 and causes prelamin A to accumulate inside the nuclear envelope, which accelerates MSC senescence. The other miRNAs, miR-543 and miR-590-3p, also indirectly control lamin A. miR-543 and miR-590-3p target AIMP3/p18 and stop MSCs from undergoing cellular senescence [126].

Potter et al. (2020) showed that miR-141 reduces an MSC’s ability to differentiate into osteogenic tissues by targeting the vitamin C transporter (SVCT2) and SDF-1. Additionally, they demonstrated that the expression of this miRNA in human and mouse bone MSCs rises with age, implying that miRNA-141 contributes significantly to MSC ageing [127].

miR-129 exhibited a positive correlation with senescence and a negative correlation with the osteogenic markers ALP, OXN, and RUNX2 in clinical bone samples. This miRNA targets Frizzled-4 (FZD4), a member of the frizzled gene family, and exerts its action through the Wnt/B-catenin pathway [128].

miR-188 is upregulated in human and old mouse BMSCs and targets MAP3K3, promoting senescence. It was well established that overexpression of miR-34a, which targets SIRT1, in MSCs causes senescence and apoptosis. Additionally, the miR-23a cluster directly targets SATB2 and controls TGF-B signaling in osteoblasts to encourage the development of terminal osteocytes. miR-21 targets E2F2, a downstream effector of p21 and p16INK4a, in breast cancer cells. Anti-TNFα therapy lowers miR-146-5p in senescent cells along with a concurrent decrease in SASP, but it has no effect on senescence levels as determined by traditional markers such as p16INK4a and SA B-galactosidase. A positive feedback loop is produced when p53 upregulates miR-34a, which in turn downregulates SIRT1, a p53 inhibitor [128].

Numerous miRNAs, including the miR-106b family, miR-130b, miR-302c, miR-302a, miR-302b, miR-302d, miR-512-3p, and miR-515-3p, have been demonstrated to protect HMECs from RasG12V-induced senescence by inhibiting the expression of p21, another crucial cell cycle regulatory protein [129].

There is a correlation between increased p16 expression and decreased miR-24 in replicative senescence. Moreover, miR-203 targets E2F3 and causes melanoma cells to entry in a senescence state [128].

#### 5.4.2. LncRNAs

LncRNAs are RNAs longer than 200 nucleotides that bind to RNA, DNA, and proteins and act as guides, enhancers, or scaffolds in post-transcriptional and post-translational regulations. The lncRNA NEAT1 might be a therapeutic target for skeletal ageing since it increases osteoclastic differentiation and stimulates CSF1 release in senescent bone marrow mesenchymal stromal cells. The lncRNA APTR accelerates the cell proliferation of primary hepatic stem cells in mice [130].

Luo et al. (2016) conducted a comparison of lncRNA expression in wild-type and DNA methylation-deficient Dnmt3a KO HSCs at different HSC ages, LncHSC-1 and LncHSC-2, which are not present in Dnmt3a KO HSCs but are substantially expressed in wild-type HSCs [127]. In aging vascular smooth muscle cells (VSMCs), Tan et al. (2019) observed an increase in the expression of miR-181a and a downregulation of ANRIL and Sirt1-AS [131]. The lncRNA ANRIL could inhibit cell cycle arrest and lower p53-/p21 pathway activity. In cervical cancer cell lines, proliferation, invasiveness, and migratory capacity have all been inhibited by ANRIL, while apoptosis and senescence have been enhanced.

GAS5 knockdown counteracted the miR-223-mediated regulation of NAMPT expression, which in turn accelerated the senescence and inhibited the proliferation of endothelial progenitor cells (EPCs). GAS5 modulates PI3K/AKT signaling to control the proliferation and senescence of EPCs. Another lncRNA that influences senescence is GUARDIN. This lncRNA is an essential component of the transcriptional repressor complex that also includes PGC1α and LRP130 [130,131].

H19 is also a long non-coding RNA that contributes to ageing. The increase in p16 and p21 in response to H19 silencing decreased cell proliferation and induced senescence in vitro. Furthermore, array data have shown that H19 is involved in IL-6 signaling. H19 targets miR-22 and determines dysregulation in cellular senescence through the Wnt signaling pathway. Another lncRNA that controls cellular senescence by adjusting miR-128 levels is HCP5 [130].

The recently discovered long intergenic RNAs (lincRNAs) such as Linc-ASEN, LINC00673, LINC00623, LINC01255, and lincRNA p21 appear to influence cellular senescence [132].

#### 5.4.3. circRNAs

It has been demonstrated that circRNA-0077930 increases VSMC senescence by upregulating the expression of KRAS, p21, p53, and p16 and decreasing the expression of miR-622 [129]. Circ-Foxo3 expression was increased in the heart tissue of old mice and humans. Circ-Foxo3 interacts with ID-1, E2F1, FAK, and HIF1α [130]. Upregulation of circLARP4 in hepatocellular carcinoma cells can cause cell cycle arrest and cellular senescence via affecting the miR-761/RUNX3 axis. The proliferative impact of let-7 has been reversed and cellular senescence has been increased through circPVT1 silencing. By suppressing circPVT1, the expression of IGF2BP1, KRAS, and HMGA2 has been reduced. Furthermore, circCCNB1 also blocks cellular senescence via influencing CCNE2 and sponging miR-449a [133]. Furthermore, circACTA2 regulates the connection between ILF3 and CDK4 and contributes to cellular senescence [130].

## 6. Animal Models of Senescence Research

Animal models are essential in senescence research, bringing a huge input in studying this inevitable biological process and the possible genetic, phenotypic, and pharmacological methods to slow it down. Aging is characterized by increasing morbidity and functional decline, representing the greatest risk factor for numerous human diseases. The unraveling of its mechanism and all the factors involved, including all epigenetic interactions, can thus facilitate the development of new treatments for diseases associated with age and, of course, for delaying this process [134,135]. Since the usefulness of animal models is limited in the study of cellular senescence, computational modeling that intervenes by increasing the value of animal models can complement it [136].

The use of humans in senescence research is complicated by many factors, primarily by ethical issues. In addition, environmental and social factors cannot be ignored. But perhaps the most pressing aspect to consider is their long natural lifespan. As almost all organisms age, animal models have become indispensable for studying this process [135].

There are many ways to approach senescence research using animal models, but in general, they can be divided into two categories: *natural aging models* and *accelerated aging models*. The main advantage of using naturally aging animal models is that they can develop many phenotypes, related to normal human aging (such as cataracts and muscle weakness). However, research using the animal model of natural aging is time-consuming, labor-intensive, and very expensive. Therefore, researchers due to its convenient source, short modeling time, and relatively controllable aging effect prefer the induced accelerated aging model [137].

A summary classification of the animal models used in senescence research groups them into invertebrates, amphibians and reptiles, fish, rodents, dogs and cats, and birds and primates.

The most frequently used animal models are rodents, such as mice (*Mus musculus*) and rats (*Rattus norvegicus domestica*), due to their well-known and obvious advantages [138] such as easy handling, short generation times, availability of standardized strains and growth, and a multitude of known data related to genomic and transcriptomic sequencing [139]. Due to their generally high longevity, African mole-rats (more than 20 years) and naked mole-rats (more than 30 years) are frequently used for the study of cellular senescence [140,141]. In mammals, longevity (expressed as the maximum lifespan of the species) is allometrically correlated with body size, but naked mole-rats (*Heterocephalus glaber*) have the highest longevity for their body size, followed by African mole-rats (*Bathyergidae*) [142]. Thus, the main reason why these species are of increased interest to researchers is that they live up to five times longer than expected, given their small body sizes [133]. In addition to increased longevity, they also present other advantages such as a very low risk of extrinsic mortality (because they live subterranean and are not exposed to natural predators or climatic effects), the present eusocial organization (the highest level of organization of sociality which brings many benefits to the individuals), and a diversity of reproductive strategies [142,143].

Naked mole-rats show signs of aging similar to those of humans, such as retinal degeneration and osteoarthritis, but show minor senescence, no signs of tumorigenesis, age-independent mortality rates, and high fecundity until death [144]. Activity levels of antioxidants, such as superoxide dismutase, do not change with age in naked mole-rats, in contrast to mice, in which they decrease with age [136]. Thus, maintenance of SOD1 and 2 activity is considered a contributing factor to their extended lifespan. In addition, the insulin/IGF signaling pathway is another important modulator of lifespan. Naked mole-rats show an unusual distribution of endocrine cells compared to most other rodents, which may explain their unusual hyperglycemic state. These animals show lower insulin levels, further highlighting the complexity of the IGF pathway in longevity [145]. Their cells produce fewer aberrant proteins, supporting the fact that the more stable is the proteome, the greater the longevity. A whole-genome sequencing analysis of naked mole-rats found that genes related to macromolecule degradation due to mitochondrially encoded genes were not altered with age. In addition, telomerase reverse transcriptase showed a stable expression regardless of age [136]. Increased longevity and amazing resistance to cancer are also due to the efficient repair of DNA damage, autophagy of damaged cells, cytoprotective pathways, telomere and epigenome maintenance, long non-coding RNAs, and a strong innate immune response [140].

Thus, improved antioxidant defense, mitonuclear balance, protein synthesis, autophagy, and increased translational fidelity are some of the particularities related to longevity that explain the enhanced interest of these animal models in research.

Due to their availability and easy handling, mice are also often used for studying cellular senescence, being especially used for research of pharmacological treatments that can extend lifespan (studies of the effect of metformin, resveratrol, rapamycin, spermidine, chloroquine) [146,147]. The best rodent model to investigate possible antiaging medications is most likely the heterogeneous mouse model (HET). For studies on aging, another mammalian rodent model used is the P8 accelerated senescence mouse [148]. APP/PS1 mice are double transgenic mice expressing a chimeric mouse/human APP (Mo/HuAPP695swe) and a mutant human presenilin 1 (PS1-dE9) and they are used especially in studying brain cell senescence. PS19 transgenic mice are used in the study of neurodegenerative pathologies (such as Alzheimer’s disease) and cognitive deficits because of the high levels of mutant human tau, specifically in neurons and appearance of gliosis, neurofibrillary tangle (NFT) deposition, neurodegeneration, and loss of cognitive function [149].

Inbred mice are most widely used for the study of senescence and age-related diseases because the genetic similarity is high, and differences can be attributed to environmental or treatment effects. The disadvantage of these animal models is represented by the fact that inbred mice, such as BALB/cJ and C57BL/6, develop a limited range of diseases. Also, differences in reported mean lifespan can vary by up to 20% depending on mouse strain and sex, despite the same genetic and phenotypic background [150]. Another disadvantage of using inbred mice in senescence research is the fact that the strain used may not be representative of the species as a whole, so the results obtained must be analyzed cautiously. Furthermore, the resulting genetic uniformity of the inbred strains is not representative for humans. Outbred mice are genetically undefined, thus having the advantage that the results obtained from their use are more representative for the human population [136].

Accelerated aging mice are a group of inbred mouse strains that are used in the study of senescence because of the particularity of accelerated aging and because they exhibit various aging-related diseases similar to humans (such as senile amyloidosis, senile osteoporosis, learning/memory impairment). In these strains, cellular senescence was identified in various cell types, including astrocytes, endothelial cells, progenitor cells, retinal epithelial cells, and fibroblasts [151]. They show an increase in ROS generated by mitochondria or other cellular sites, which not only causes damage to mitochondria but also triggers the degradation that leads to the outcome of aging [138].

Concerning the use of mice as animal models, regardless of the used strains, it allows researchers to carry out the study in a very short time, controlling the environmental factors that influence aging, and allows them to examine in detail the genetics and mechanisms involved in this process [152].

Domesticated species such as dogs and cats can represent valuable animal models in the study of cellular senescence because they develop many pathologies associated with age, but the average lifespan discourages longevity studies [136].

Due to the existing similarities, especially for the mechanisms of aging and associated diseases, primates represent important animal models for the study of cellular senescence. The main species used in aging studies are Rhesus macaques (*Macaca mulatta*), chimpanzees (*Pan troglodytes*), bonobos (*P. paniscus*), the white-faced capuchin monkey (*Cebus capucinus*), and the common marmoset (*Callithrix jacchus*). But there are many disadvantages that reduce the frequency of their use as animal models, among which are the difficulty of growing (due to their aggressiveness, strength and weight), special considerations (due to their intelligence), the possibility of transporting and transmitting numerous pathogens, ethical concerns, and high financial costs [136,138].

The use of fish, reptiles, amphibians or invertebrates as animal models in senescence studies is less common but, as in the case of rodents, it suggests a similar positive allometric relationship of body weight and lifespan [138].

Invertebrate models are mainly used for drug screening in the study of cellular senescence [146]. The common fruit fly (*Drosophila melanogaster*) and roundworm (*Caenorhabditis elegans*) are often used due to short generation times for aging research [138,153]. *Caenorhabditis elegans* (*C. elegans*) is a nematode especially used as an animal model for research involving cellular senescence, aging, and neurodegenerative diseases such as Alzheimer’s and Parkinson’s. Compared with traditional animal models, this nematode has many advantages, such as small body size, short lifespan, complete sequenced genome, and more than 65% of genes associated with human diseases. All these benefits make this organism an ideal living system for senescence research [152].

For the study of cellular senescence and aging, the species of the Hydra genus are of particular interest due to their remarkable cellular regeneration potential and longevity. The regeneration capacity is due to the high proportion of stem cells, which ensures constant self-renewal with a cell cycle of one to four days. Similar to Hydra, planarian flatworms (class Turbellaria) are described as having an apparent biological immortality due to their ability to regenerate [139].

Among the species of marine invertebrates, species that have a very long lifespan can be of interest for research in the field of senescence, such as the ocean quahog *(Arctica islandica*) that can live up to 500 years, the red sea urchin (*Strongylocentrotus franciscanus*), which reaches a lifespan of about 200 years or *Homarus* which can live up to 50 years in the wild and up to 100 years in captivity [139].

Among the amphibians used in the study of cellular senescence are the neotenic salamander (*Proteus anguinus*), which has an average lifespan of 68 years [154]; the common mudpuppy (*Necturus maculosus*), which has an estimated lifespan of 34 years [155]; and the axolotl (*Ambystoma mexicanum*), which lives about 20 years [156]. These species stand out, in addition to the increased longevity in relation to their body size, by an extremely low incidence of cancer and a unique capacity for regeneration (having the ability to regenerate certain wounded tissues, the tail, or even limbs) [140].

Among the reptiles, sea turtles that live over 100 years are used for aging research, for example, giant turtles. The disadvantage of using them as animal models is their size and the challenges of maintaining captive breeding colonies [157]. Long-term monitoring of wild turtle populations has shown that species in the *Cheloniidae* family have several adaptations to extensive hypoxia that may have contributed to their resistance to age-related oxidative damage [140].

Fish are used as animal models mainly due to the ease of cultivation and growth, the most common fish used in the laboratory being the Zebrafish (*Danio rerio*). The fish species present in cellular senescence studies are *Nothbranchius furzeri*, which has a short lifespan (~13 weeks) but is easy to reproduce and can be kept at relatively high population densities, allowing for rapid expansion, thus providing an easy and inexpensive animal model [136,158]. Among the long-lived Fishes, the Rockfishes are used because they present a variation in and a large number of species, and in their case, it was discovered that there was a positive correlation between depth and longevity, and the longest-lived species do not seem to be subject to reproductive senescence [140,159].

Unlike mammals and invertebrates, where longevity increases in relation to body size, birds have a long life compared to their small size. Possible explanations for their increased lifespan are decreased susceptibility to oxidative stress, increased telomere length [160], and peculiarities of the insulin/IGF-1 pathway (birds maintain blood glucose levels one to three times higher than most mammals, but with low insulin and high glucagon levels) [161,162]. The advantages of using birds as animal models are represented by their ability to reproduce, maintaining fecundity with age and ease of growth in the laboratory environment. The main disadvantage of this animal model is the substantial differences between the physiological and biological mechanisms of birds and mammals [136,138].

All animal models used in cellular senescence research have advantages and disadvantages, but it is necessary to balance the benefits of using them and their limitations in choosing the right animal model for each individual study. Animal models are the basis for biomedical research but the challenges for the future will be the extrapolation of the information obtained from these animals to humans and their use to understand all the factors and mechanisms involved in cellular senescence. Also, cancer stem cells are a good model for studying the origin of different neoplasms. Cancer stem cells are a small subpopulation of tumor-forming cells that promote carcinogenesis and metastasis and confer tumor heterogeneity. Cancer stem cells are defined by their self-renewal and continuous proliferation. Furthermore, these cells have superior ability to penetrate, invade, multiply, migrate, initiate tumor growth, and to create phenotypes that become resistant to different drugs [162,163,164,165,166].

## 7. Conclusions and Future Perspectives

Diet and exercise are seen to be the best and most straightforward ways to postpone senescence, while medications that target important cellular and molecular changes associated with senescence remaining as intriguing topics for studying aging-related disorders. Various comparable treatment techniques have been developed based on these epigenetic changes in cells throughout senescence. In preclinical research, drugs like metformin, rapamycin, dasatinib, and other medications have also demonstrated beneficial effects in reducing diseases associated with aging and controlling aging-related epigenetic modifications. The development of senolytic vaccines was meant to target both senescent cells in tissues and senescent immune cells. Also, other types of senolytic vaccines are those that target the release of antiaging molecules and antiaging factors. Although there are clinical trials involving these senolytics and their role in lowering the number of senescent cells in humans, the short-term and long-term side-effects of these therapies are still unknown. Reprogramming of somatic senescent cells using Yamanaka factors (Oct3/4, Sox2, Klf4, and c-Myc; OSKM) to produce induced pluripotent stem cells using both transient and long-term methods may offer a novel way to slow down an organism’s aging process in vivo. Even with all the recent advancements, it is still unknown how epigenetic modifications interact with genetic background and even how the microbiota controls the aging process. Furthermore, the roles that lifestyle choices, physiological conditions, and psychological states play in the epigenetic modifications associated with aging remain unknown.

## Figures and Tables

**Figure 1 ijms-25-09708-f001:**
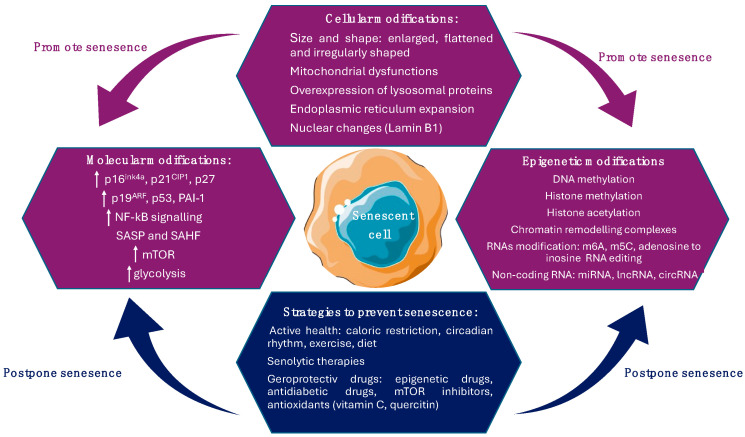
Cellular and molecular modification that promote senescence and strategies aim to alleviate senescence.

**Figure 2 ijms-25-09708-f002:**
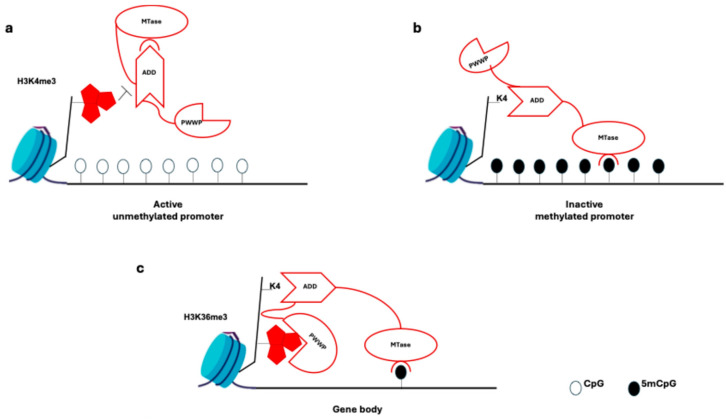
DNA methylation mechanism at the promoter level. (**a**) DNA methylation at promoters: trimethylated histone H3 Lys4 (H3K4me3) prevents binding to chromatin of the ADD domain of DNMT3A and DNMT3B (and of DNMT3L). These interactions cause it to bind to the methyltransferase (MTase) domain and auto-inhibit the DNMT3 enzymes. (**b**) In the absence of H3K4 methylation, the ADD domain binds to H3K4 and the auto-inhibition is relieved, thereby allowing the MTase domain to methylate the DNA. (**c**) In gene bodies, the ADD domain binds unmethylated H3K4, thereby releasing the auto-inhibition of the DNMT3 enzymes. H3K36me3 is deposited in the gene bodies of actively transcribed genes and serves as a recruitment module for the DNMT3 PWWP domain.

**Figure 3 ijms-25-09708-f003:**
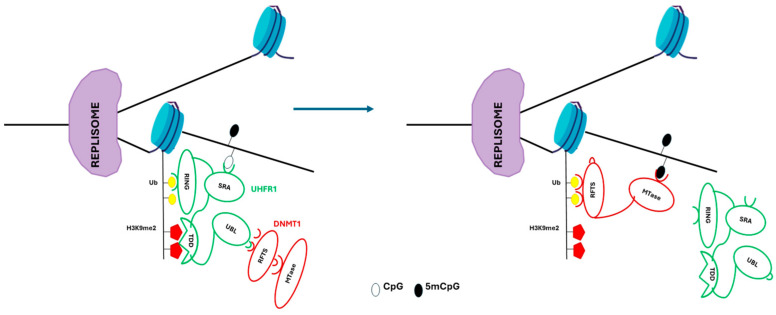
DNA methylation mechanism at replication level. UHRF1 is recruited to replicate DNA through SRA domain, and binds hemimethylated CpG sites. The TTD domain binds H3K9me2. The RING domain of UHRF1 ubiquitylates the histone H3 (Ub). The replication foci-targeting sequence (RFTS) of DNMT1 folds into the MTase domain, thereby preventing its catalytic activity. UHRF1 enrolls DNMT1 through an interaction between its ubiquitin-like (UBL) domain and the DNMT1 RFTS. Auto-inhibition of DNMT1 is released when the RFTS binds to ubiquitylated H3 tails, which enables the maintenance of symmetrical DNA methylation at CpG sites.

**Table 1 ijms-25-09708-t001:** DNA methylation and demethylation enzyme and factors and their role [64,65,66,67,68,69].

Enzyme/Factor	Role	Mouse Loss-of-Function Model	Human Diseases Due to Mutations
**DNMT3A**	De novo DNA methyltransferase	Knockout mice die 4 weeks after birth In germline-specific knockout sterility in both males and females	Microcephalic dwarfism Tatton-Brown–Rahman syndrome Acute myeloid leukemia
**DNMT3B**	De novo DNA methyltransferase	Dnmt3b–/– double mutants exhibit early embryonic lethality	Immunodeficiency centromeric instability and facial anomalies syndrome
**DNMT3C**	De novo DNA methyltransferase (mice/rats specific isoform)	Infertile males due to defect in methylating transposon promoters during spermatogenesis	
**DNMT3L**	DNA methyltransferase cofactor	Meiosis is not possible for male germline cells. Mid-gestation lethality	
**DNMT1**	Maintenance DNA methyltransferase	Embryonic lethality Low DNA methylation Derepression of transposons	Autosomal dominant cerebellar ataxia, deafness, and narcolepsy Hereditary sensory autonomic neuropathy 1E
**UHFR1**	DNMT1 cofactor	Embryonic lethality Low DNA methylation	
**TET1**	DNA demethylation via oxidation of methylcytosine	Tet1–/–, Tet2–/– double mutants: developmental defects and partial lethality	
**TET2**	DNA demethylation via oxidation of methylcytosine	Increased self-renewal of hematopoietic stem cells	Acute myeloid leukemia Chronic myelomonocytic leukemia Lymphomas Myeloproliferative neoplasms
**TET3**	DNA demethylation via oxidation of methylcytosine	Impaired paternal demethylation Reduced fecundity	

**Table 2 ijms-25-09708-t002:** Classification of HDAC and HAT enzymes and their inhibitors [103,104,105,106,107].

Histone Deacetylase				
Classification			Localization	Inhibitors (Examples)
**Zn+** **dependent**	Class I	HDAC 1 HDAC 2 HDAC 3 HDAC 8	Mainly nucleus Nucleus Nucleus/cytoplasm Mainly cytoplasm	**Benzamides** (MS-275, MCGD0103, CI-994) **Cyclic peptide** (Depsipeptide, Apicidin) **Aliphatic fatty acids** (butyrate, valproic acid) **Hydroxamate** (SAHA, PXD100, LBH589, 4SC-201, Tubacin, ITF2357, PCI2478I) **Mercaptoketone** (KD5170)
	Class II	HDAC 4 HDAC 5 HDAC 6 HDAC 7 HDAC 9 HDAC 10	Nucleus/cytoplasm Nucleus/cytoplasm Cytoplasm Nucleus/cytoplasm/ mitochondria Nucleus/cytoplasm Nucleus/cytoplasm	**Aliphatic fatty acids** (butyrate, valproic acid) **Hydroxamate** (SAHA, PXD100, LBH589, 4SC-201, Tubacin, ITF2357, PCI2478I) **Mercaptoketone** (KD5170)
	Class IV	HDAC 11	Mainly nucleus	**Hydroxamate** (SAHA, PXD100, IFT2357, 4SC-201)
NAD+ dependent	Class III	SIRT 1 SIRT 2 SIRT 3 SIRT 4 SIRT 5 SIRT 6 SIRT 7	Cytoplasm Cytoplasm/nucleus Mitochondria Mitochondria Mitochondria Nucleus Nucleus	**Hydroxamate** (SAHA, PXD100, IFT2357, 4SC-201) **Benzamides** (MCGD0103
**Histone Acetyltransferase**				
**Classification**			**Localization**	**Inhibitors** **(Eexamples, * Nonselective)**
Zn+ dependent	**Cytoplasmic**	KAT1 (HAT1) HAT4 (NAA60) HAT2 HATB3.1 Rtt109	Cytoplasm	*Anacardic acid *Isothiazolones
	**GNAT (bromodomain)**	KAT2A (Gcn5) KAT2B (PCAF) ELP3	Nucleus	Ischemin Ischemin *Anacardic acid *Isothiazolones
	**MYST (acetyl-CoA motif)**	KAT5 (TIP60) KAT6A (MOZ, MYST3) KAT6B (MORF, MYST4) KAT7 (HBO1, MYST2) KAT8 (MOF, MYST1)	Nucleus	TH1834 *Anacardic acid *Isothiazolones
NAD+ dependent	**P300/CBP**	KAT3B (p300) KAT3A (CBP)	Nucleus	Garcinol, curcumin, benzylidene barbituric acid, C646, CTPB, TTk21 TTK21, ICG-001, ischemin
	**Transcription** **co-activators**	KAT4 (TAF1, TBP) KAT12 (TIFIIIC90)	Nucleus	*Anacardic acid *Isothiazolones
	**Steroid receptor** **co-activators**	KAT13A (SRC1) KAT13B (SCR3, ACTR) KAT13C (p600) KAT13D (CLOCK)	Nucleus	*Anacardic acid *Isothiazolones

**Table 3 ijms-25-09708-t003:** Epigenetic modification and their role in stem cell senescence [59,60,61,62,63,64,65,66,67,68,69,70,71,72,73,74,75,76,77,78,79,80,81,82,83,84,85,86,87,88,89,90,91,92,93,94,95,96,97,98,99,100,101,102,103,104,105,106,107,108,109,110,111,112,113,114,115,116,117,118,119,120,121,122,123,124,125,126].

Epigenetic Modification	Substrate	Enzymes/Target	Senescence
DNA methylation DNA demethylation	CpG sites	DNMT3A, DNMT3B, DMNT1	↑
		TET1, TET2, TET3	↓
Histone methylation	H3K4me3	KDM5A, ASH1L	↑
	H3K9m2/3	JMJD-1.2/PHF8, JMJD-3.1/JMJD3	↓
	H3K23me3	JMJD-1.2/PHF8, JMJD-3.1/JMJD3	↓
	H3K27me3	JMJD-1.2/PHF8, JMJD-3.1/JMJD3	↓
	H3K27me3	EZH2	↑
	H3K36me2/3	SETD2	↑
	H3R2/17/26me	CARM1data	↑
Histone acetylation	H3K9ac	SIRT6	↓
	H3K56ac	SIRT6	↓
	K3K27ac	HDAC4	↑
	H3K14ac	KAT7	↑
	H3K9ac	GCN5 (KAT2A)	↑
	H3K9ac	PCAF (KAT2B)	↑
Histone phosphorylation	H3S10p	Aurora kinase	↑
	H3T11p	CK2	↑
	H3S28p	in Drosophila cells	↓
	γH2AX	ATM, ATR	↑
Histone ubiquitination	H2Bub	BRCA1/BARD1; E3 ligase	↑
	H2Aub	BRCA1/BARD1; E3 ligase	↑
Chromatin remodeling complex	SWI/SNF	ARID1B/ENTPD7	↑
	NuRD	HDAC1	↑
RNA modifications	m6A m5C A-to-I RNA editing	methyltransferases: METTL3/14/16, RBM15/15B, ZC3H3, VIRMA, CBLL1, WTAP, and KIAA1429 demethylases: FTO and ALKBH5 tRNA methyltransferase NSUN2 ADAR1, ADAR2	↑ ↑ ↑ ↓
miRNA	mir17 family miR-195 miR-486-5p, miR-204 miR-495 miR-141-3p, miR-543, miR-590-3p miR-543, miR-590-3p miR-141 osteogenic markers miR-188 miR-21 miR-146-5p miR-34a miR-106b family, miR-130b, miR-302a, miR-302b, miR-302c, miR-302d, miR-512-3p, and miR-515-3p	p12 SIRT1, TERT BMI1 ZMPSTE24 AIMP3/p18 SVCT2, DF-1 Frizzled-4 MAP3K3 E2F2 TNFα p53 p21	↓ ↓ ↑ ↑ ↑ ↓ ↑ ↑ ↑ ↑ ↑ ↑ ↓
LncRNA	NEAT1 APTR LncHSC-1, LncHSC-2 ANRIL GAS5 GUARDIN H19 HCP5	CSF1 Dnmt3a p53 -p21 pathway NAMPT, PI3K/AKT pathway PGC1α and LRP130 p16 -p21 pathway, miR-22 miR-128	↑ ↓ ↓ ↓ ↑ ↑ ↓ ↑
circRNA	circRNA-0077930	KRAS, p21, p53, p16, miR-622	↑
	circ-Foxo3	ID-1, E2F1, FAK, HIF1α	↑
	circLARP4	miR-761/RUNX3 axis	↑
	circPVT1	let-7	↓
	circCCNB1	CCNE2, miR-449a	↓
	circACTA2	ILF3, CDK4	↑
▬ methyltransferase ▬ demethylase ▬ histone deacetylase ▬ histone acetyltransferase			

## Data Availability

Not applicable.

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
