# Peer review of "Genetic and Epigenetic Interactions Involved in Senescence of Stem Cells"

_ijms, 2024, doi:10.3390/ijms25179708_

Round 1

Reviewer 1 Report

Comments and Suggestions for Authors

The work by Iordache et al, entitled 'Genetic and epigenetic interactions involved in senescence of stem cells', raises an important topic. Consideration of molecular mechanisms of stem cell aging contributes to defining new directions for therapeutic use of stem cells. The authors describe molecular mechanisms of aging in detail, but I have a few comments on the manuscript:

1. All statements should be provided with citations, in the work there are many places where citations are missing, for example subsection 4.4 lines 543-550, section 5, subsection 5.1 lines 5882-618 and others.

2. In tables 2 and 3, citations should be supplemented.

3. Take care of the coherence of information, in many places in the text individual fragments seem to be "detached" from the rest, take care of a smooth transition from one topic to the next.

The work should be supplemented with several issues:

- Maybe it is worth briefly noting the differences in the molecular mechanisms of aging of stem and differentiated cells?

- As stem cells age, the paracrine signaling of stem cells changes, and thus the epigenetic factors secreted by them, it is worth mentioning

- What is the impact on the molecular mechanisms of cell aging of the niche/aging of the stem cell niche, aging of the stem cell microenvironment?

- How do the discussed mechanisms of stem cell aging translate into possible interventions/rejuvenation strategies, what should future research directions focus on?

Author Response

We appreciate the time and effort that you have dedicated to providing your valuable feedback on our paper that will improve the quality of the manuscript.

  1. All statements should be provided with citations, in the work there are many places where citations are missing, for example subsection 4.4 lines 543-550, section 5, subsection 5.1 lines 5882-618 and others.

Thank you for this observation, we add the citations for these sections and check the manuscript.

  1. In tables 2 and 3, citations should be supplemented.

We add the citations for table 2 and 3.

  1. Take care of the coherence of information, in many places in the text individual fragments seem to be "detached" from the rest, take care of a smooth transition from one topic to the next.

We reread the manuscript and made a smooth transition between the fragments.

 The work should be supplemented with several issues:

- Maybe it is worth briefly noting the differences in the molecular mechanisms of aging of stem and differentiated cells?

Thank you for pointing this out. We added this paragraph regarding the differences in the molecular mechanisms of aging of stem and differentiated cells.

Stem cells are not impacted by replicative senescence, although they are nonetheless prone to harm and proliferate during the ageing process. Stem cells tend to accumulate DNA damage with ageing, which diminishes their capacity to regenerate cell lineages, resulting in age-related loss of organ function and homeostasis as well as an increase in the prevalence of age-related disorders. Nevertheless, there are not many data about what causes this damage or how it contributes to the senescence of stem cells. DNA damage can cause stem cell numbers to decrease by activating apoptosis, ageing, and differentiation pathways. Research studies has also demonstrated that aberrant cell proliferation, tumor-like alterations, and reduced self-replication of stem cells are caused by elevated ROS levels in ageing mesenchymal stem cells and elevated ROS expression in mouse haematopoietic and neural stem cells. Also, autophagy dysregulation results in abnormalities related to protein homeostasis, poor protein folding, and the build-up of abnormal proteins, which can cause damage to cells and tissues as well as deplete stem cells. Reduced stem cell uptake of nutrients is caused by an increase in point mutations and deletions in the mitochondrial DNA. The regulation of stem cell function is significantly influenced by epigenetic regulation, and alterations in the epigenome with ageing impact the stem cell ageing process. DNA methyltransferases regulate the balance between differentiation and self-renewal in many adult stem cell compartments. Genetic material damage, noncoding RNA and exosomes, loss of proteostasis, intracellular signaling pathways, and mitochondrial malfunction are the hallmarks of the molecular mechanisms of stem cell ageing. A marker of cellular senescence p16INK4A accumulates with age in NSCs, HSCs, and satellite cells.

The senescent cells found in tissues and organs play a crucial role in both the emergence of chronic diseases and the ageing process. Called cellular senescence, ageing causes cells subjected to metabolic, genotoxic, or oncogene-induced stress to enter a nearly irreversible cell cycle stop. An increase in inflammatory mediators, primarily cytokines and chemokines, known as the SASP, is a prominent characteristic of senescent cells and is hypothesised to contribute to disease. This disruption of stem cell regeneration, tissue and wound repair, and inflammation results in harm to dynamic equilibrium.

- As stem cells age, the paracrine signaling of stem cells changes, and thus the epigenetic factors secreted by them, it is worth mentioning.

Thank you for this observation. We add a paragraph regarding the paracrine signaling of stem cells.

Through the secretion of cytokines and chemokines (e.g., IL1α, IL1β, IL6, IL8, CXCL1, CXCL2), growth factors (e.g., amphiregulin, EGF, BMPs, FGFs, VEGF, WNTs), extracellular matrix components (e.g., fibronectin) and proteases (e.g., MMPs, plasminogen activators), as well as exosome-like small extracellular vesicles, the SASP mediates the paracrine activities of senescent cells. Cells that have aged or experienced senescence release many extracellular vehicles (EVs). Apoptotic bodies, microvesicles, and exosomes are examples of EVs that facilitate material transfer and cell-to-cell contact. MSCs release exosomes with modified payloads, like miRNAs, in large quantities as they age.  SASPs can trigger the proliferation, angiogenesis, or epithelial-mesenchymal transition of neighboring or cancer cells. 

- How do the discussed mechanisms of stem cell aging translate into possible interventions/rejuvenation strategies, what should future research directions focus on?

We highlighted this issue in the introduction and the conclusion part where we talked about different classes of drugs and vaccines that can target these molecules and what are the newly strategies to prevent senescence.

Reviewer 2 Report

Comments and Suggestions for Authors

This is a well written and comprehensive review.

I have only a few comments:

- Introduction: line 34-40: Is premature senescence identic to therapy-induced senescence (TIS)? The term TIS is often used in connection with anticancer drugs, which are treated in in this paragraph. I propose to clarify the terms.

- I also propose to add a note indicating that in some cancers, like glioblastoma, therapeutics can induce more senescence than cell death, indicating that the senescence pathways can easily be evoked, which has significant therapeutic implictions (see Beltzig et al., Senescence is the main trait ..., Cancers, 2022, https://doi.org/10.3390/cancers14092233).

- Line 81: It is correct that ROS can cause apoptosis. However, a hallmark of senescent cells is apoptosis resistance following ROS and other genotoxic exposures. This is why we do need senolytics. Please complete by mentioning apoptosis resistance of senescent cells.

- Finally, it is not clear how stem cells (e.g. cancer stem cells) differ in their responses from cancer non-stem cells. What makes cancer stem cells unique? I propose to add a short paragraph at the end of the manuscript.

Thanks.

Author Response

We appreciate the time and effort that you have dedicated to providing your valuable feedback on our paper that will improve the quality of the manuscript.

This is a well written and comprehensive review.

I have only a few comments:

- Introduction: line 34-40: Is premature senescence identic to therapy-induced senescence (TIS)? The term TIS is often used in connection with anticancer drugs, which are treated in in this paragraph. I propose to clarify the terms.

Thank you for pointing this out. We clarified this aspect.

- I also propose to add a note indicating that in some cancers, like glioblastoma, therapeutics can induce more senescence than cell death, indicating that the senescence pathways can easily be evoked, which has significant therapeutic implications (see Beltzig et al., Senescence is the main trait ..., Cancers, 2022, https://doi.org/10.3390/cancers14092233).

Thank you for this observation. We add this statement and the reference.

- Line 81: It is correct that ROS can cause apoptosis. However, a hallmark of senescent cells is apoptosis resistance following ROS and other genotoxic exposures. This is why we do need senolytics. Please complete by mentioning apoptosis resistance of senescent cells.

Thank you for completing the paragraph. We add this statement.

- Finally, it is not clear how stem cells (e.g. cancer stem cells) differ in their responses from cancer non-stem cells. What makes cancer stem cells unique? I propose to add a short paragraph at the end of the manuscript.

Thank you for this observation. We add a short paragraph at the end of the manuscript.

Round 2

Reviewer 1 Report

Comments and Suggestions for Authors

The authors followed the proposed comments, however in several places entire paragraphs are not provided with citations, for example: lines 471-477, 538-540, 642-648, 686-688, 711-714, 754-758.

Author Response

We appreciate very much the time and effort that you give for reviewing this manuscript.  Your valuable feedback on our paper will improve the quality of the manuscript.

The authors followed the proposed comments, however in several places entire paragraphs are not provided with citations, for example: lines 471-477, 538-540, 642-648, 686-688, 711-714, 754-758.

Thank you for these observations, we add the citations for these sections and check the manuscript.